# Potential of Using Waste Materials in Flexible Pavement Structures Identified by Optimization Design Approach

**Primož Jelušič** [1], **Süleyman Gücek** [2], **Bojan Žlender** [1,*], **Cahit Gürer** [2], **Rok Varga** [1], **Tamara Bračko** [1], **Murat V. Taciroğlu** [3], **Burak E. Korkmaz** [2], **Şule Yarcı** [2] and **Borut Macuh** [1]

[1] Faculty of Civil Engineering, Transportation Engineering and Architecture, University of Maribor, Smetanova 17, 2000 Maribor, Slovenia; primoz.jelusic@um.si (P.J.); rok.varga@um.si (R.V.); tamara.bracko@um.si (T.B.); borut.macuh@um.si (B.M.)

[2] Faculty of Engineering, Afyon Kocatepe University, Afyonkarahisar 03200, Turkey; sgucek@aku.edu.tr (S.G.); cgurer@aku.edu.tr (C.G.); eniskorkmaz@aku.edu.tr (B.E.K.); syarci@aku.edu.tr (Ş.Y.)

[3] Faculty of Engineering, Department of Civil Engineering, Mersin University, Mersin 33343, Turkey; mtaciroglu@mersin.edu.tr

\* Correspondence: bojan.zlender@um.si

**Abstract:** This paper presents the design of geosynthetic reinforced flexible pavements and their modification by incorporating waste materials into bonded and unbonded layers of the pavement structure. The optimal design of flexible pavements was achieved by minimizing the construction cost of the pavement. The incorporation of waste materials into the pavement structure affects the material properties. Therefore, along with the traffic load, the effects of the material properties of the asphalt concrete, base layer, sub-base layer, and subgrade were analyzed in terms of pavement structure costs and $CO_2$ emissions of materials used in pavement construction. In addition, a comparison was made between pavements with and without geosynthetic reinforcement in terms of design, optimum construction cost, and $CO_2$ emissions. The use of geosynthetics is even more effective in pavement structures that contain waste materials in an unbound layer, both in terms of cost and $CO_2$ emissions. The minimum value of the California Bearing Ratio of the subgrade was determined at which the use of geosynthetic reinforcement for pavement structure with and without the inclusion of waste materials is economically and sustainably justified. The use of geosynthetics could result in a 15% reduction in pavement structure cost and a 9% reduction in $CO_2$ emissions due to the reduced thickness of unbound layers. In addition, reducing the CBR of the unbound layer from 100% to 30% due to the inclusion of waste materials implies a cost increase of up to 13%. While the present study is based on an empirical pavement design method in which pavement thickness is limited by the pavement thickness index, the same minimum thicknesses are obtained in the optimization process regardless of whether the objective function is the minimum construction cost or minimum $CO_2$ emissions.

**Keywords:** pavement design; waste materials; optimization; minimum construction cost; $CO_2$ emissions; geosynthetics

## 1. Introduction

The construction of buildings, roads, railroads, power grids, and other infrastructure often generates large amounts of clean and contaminated waste, most of which ends up in landfills. Therefore, measures that focus on sustainable management are important. Smarter methods are needed to reduce waste and ensure that its reuse does not pose a risk to public health or the environment. An analysis of current waste reuse practices has identified the main barriers (legal, organizational, logistical, and material quality) to effective reuse. The (re)use of waste, including excavated contaminated or uncontaminated soils, offers the following benefits [1]:

- Reduction in costs associated with disposal.
- Preservation of landfill capacity.
- Conservation of mined natural resources.
- Reduction in environmental and ecological impacts.

Waste management plays a critical role in road construction and encompasses a number of important aspects, as pointed out by Hale et al. [1]. First and foremost, a legal framework ensures that construction projects comply with environmental regulations and waste disposal laws to minimize potential negative impacts on the ecosystem. The organizational aspect involves effective planning and coordination among stakeholders to promote efficient waste sorting, collection, and disposal processes. Logistical and economic considerations optimize resource allocation, reduce costs, and minimize the environmental footprint associated with waste transportation and disposal. In addition, maintaining material quality is critical to ensure that waste materials incorporated into road construction meet required standards and improve the overall infrastructure durability and performance. Convergence of these aspects is essential to ensure that waste is not simply disposed of, but reused through appropriate management practices, contributing to sustainable road construction practices.

In the design of road pavements, many of the technical conditions set out in standards and technical specifications must be met to ensure the quality of the material, which is difficult to achieve when using waste. Waste may be contaminated depending on its origin, so its level of contamination must be well characterized and evaluated as a first step for reuse. In addition, these wastes may have quality differences and therefore need to be characterized in order to assess whether the waste in question can be reused for the proposed purpose (Figure 1). For this reason, several research works have been carried out to precisely determine the performance of the material containing the different wastes. Huang et al. [2] studied the different types of waste materials that can be used in road pavements. The study included waste glass, steel slag, tires, and plastics, and investigated the wet and dry methods, in which the waste material is mixed with bitumen, and the dry method, in which the waste material replaces the fine aggregates in asphalt. The study concluded that waste materials could prove to be valuable substitutes in the construction process. Similar studies were conducted and confirmed the statement that waste materials can be a valuable substitute for virgin material [3–5]. The properties of subgrade when plastic waste is added were also analyzed. Abukhettala and Fall [6] concluded that the amount of plastic waste added and the change in the CBR value of the subgrade is not linearly correlated, but that depending on the type and form of plastic waste added, there is a critical value beyond which the CBR value decreases. This study also suggests that this is related to the reinforcing capacity of the plastic waste, as a smaller amount of plastic is more easily distributed in the subgrade and therefore has a greater chance of increasing the CBR value of the subgrade. Subgrade in particular clay was also reinforced using fly ash. Studies showed a significant improvement in the CBR value of the subgrade when 15% fly ash was incorporated into the subgrade layer, reducing the required thickness and lowering the cost of the asphalt pavement structure [7,8]. To determine the foundation properties of bitumen containing plastic waste, models have been developed using various machine learning algorithms [9]. Machine learning algorithms have also been applied to predict the Marshall Stability of asphalt concrete based on bitumen content, plastic content, bitumen grade, and plastic size [10]. Lee and Le [11] quantitatively investigated the effects of adding waste plastic aggregates and magnesium-based additives in asphalt mixtures and evaluated various properties such as the deformation strength, indirect tensile strength, rut depth, and dynamic stability.

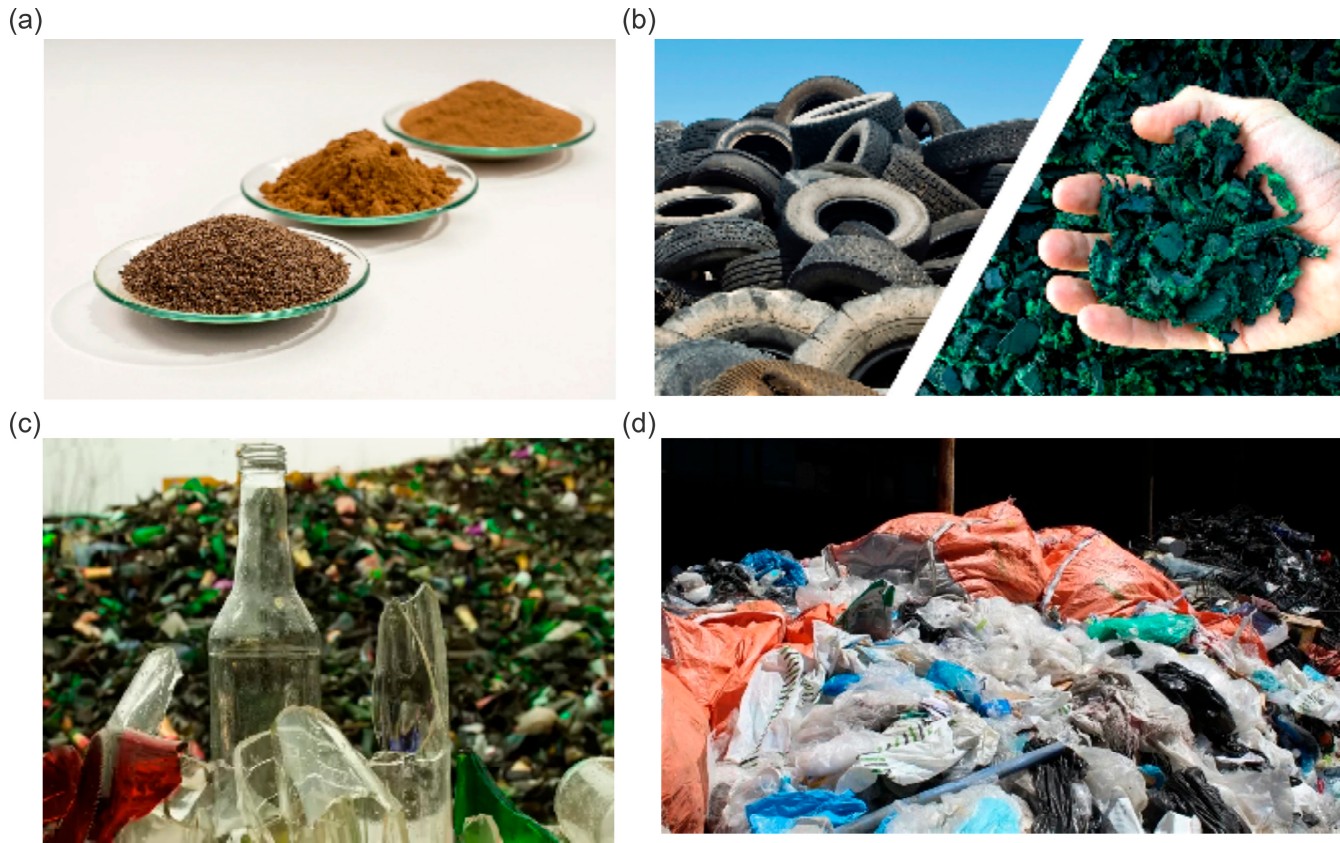

**Figure 1.** Typical types of waste: (**a**) Lignin, (**b**) car tire waste, (**c**) glass, and (**d**) nylon bags.

Because waste materials have been shown to be useful and enhance performance or be on par with virgin materials, various studies have been conducted on the economic and environmental impacts that using waste material has over virgin materials. The cost of asphalt pavement is mainly caused by the asphalt itself. Approximately 60% of the asphalt cost is bitumen, so the main objective is to reduce the bitumen content to significantly affect the cost savings [12]. Therefore, studies have been conducted on the usage of waste plastic in bitumen to save costs. Vasudevan et al. [13] concluded that using waste plastic is an ecofriendly disposal of plastic and that the performance of these roads is on par with roads constructed with virgin materials while the costs were also reduced. The same conclusions were also documented by several authors for the usage of crumb rubber to also reduce the pavement costs for the same or even better performance [14,15].

Furthermore, following the costs, the environmental impact of using waste materials has also been documented by several authors. To estimate carbon emissions from recycled construction waste and other materials, a grey model that can be used in various situations to estimate carbon emissions from recycling activities and carbon emissions from alternative materials can benefit the environment. Wang et al. [16] proved the environmental benefits of recycling and the use of waste materials in construction. Mechanical and environmental performance was also studied and showed that including slag in asphalt mixes instead of basalt aggregates showed environmental and cost advantages while also providing better mechanical performance of the asphalt pavement structure while under load [17]. You et al. [18] studied the integration of recycled plastic waste into asphalt pavements. This study showed that using recycled plastic waste in asphalt pavements instead of the original plastic and applying the right process to reduce the generation of toxic exhaust gasses that further pollute the climate can prove to be beneficial to the environment. It should be noted, however, that despite some improvement in rutting, aging resistance, and tensile strength of asphalt pavements, fatigue life and resistance to cracking at low temperatures are significantly lower when plastic waste is used. White et al. [19] considered

climate change and proposed a method for estimating $CO_2$ emissions and the cost of using alternative materials for road construction. LCA research of bituminous mixtures containing recycled materials such as crumb rubber found significant environmental impact and energy savings benefits when wet technology was used, but showed almost no benefits when the dry technology was used, which the authors attributed to the lack of data on the maintenance and life cycle of rubber-reinforced asphalt [20]. Practical applications of the use of plastic waste have shown that it is possible to build durable roads from plastic and that the construction of such roads reduces $CO_2$ emissions compared to conventional road pavements [21–23]. The research suggests that waste material is an important building stone in pavement design and should be considered when constructing a new pavement. This paper further evaluates the impact of the quality of the waste material and virgin material mixture on the optimal pavement design and its $CO_2$ equivalent.

Methods for designing pavements vary, with some being purely empirical and others mechanistic-empirical. Base course thickness can be reduced with geosynthetics. The performance of geosynthetic-reinforced flexible pavements can be evaluated using field tests, laboratory tests, and numerical simulations. To compare conventional flexible pavements with geosynthetic-reinforced flexible pavements, both unreinforced and reinforced pavement types should be optimized at a minimum cost. For this purpose, optimization models were developed and advanced algorithms were applied [24].

The main objective of this study is to investigate the feasibility of integrating waste materials into bound and unbound layers of the road pavement while evaluating the impact of these waste materials on the overall pavement design. This investigation relies on the utilization of existing empirical methods for pavement design in conjunction with an in-depth analysis of research efforts aimed at describing the material properties of these waste constituents. Such an approach considers the complex interplay between the inclusion of waste materials and the impact on the economic viability and environmental sustainability of road pavements. Exploration of this multifaceted area seeks to unravel the intricate link between waste utilization, pavement design optimization, and the broader considerations of cost-effectiveness and ecological responsibility. A genetic algorithm was employed in this study to optimize pavement structure cost and $CO_2$ emissions in accordance with pavement design guidelines. Genetic algorithms have proven to be very effective because they do not require differentiable functions, which is advantageous for complex nonlinear functions such as those often used in pavement optimization. The main novelty of this work is the optimization model, which allows us to obtain an optimal design of the pavement structure and is able to take into account different material properties affected by the inclusion of waste materials, both in terms of cost and $CO_2$ emissions. Such an approach demonstrates the potential of using waste materials in flexible pavements in terms of cost and $CO_2$ emissions. To represent the effects of the material properties of the asphalt, base layer, sub-base layer, and subgrade, all of which are affected by the inclusion of waste materials, a parametric study was conducted to determine the optimal design of flexible pavement structures using the developed optimization model. Due to the large number of combinations of design parameters (material properties, traffic loads, and geosynthetic reinforcement) involved in the optimization process, manual execution of the algorithm was not possible. Therefore, an optimization model was developed that includes a loop that performs optimizations for each combination. In addition, a sensitivity analysis was performed on the importance of each design parameter based on all optimal solutions.

## 2. The Performance of Asphalt and Unbound Layers Containing Waste Materials

The performance of asphalt mixtures containing waste materials has been investigated in Marshall tests, and numerous studies have concluded that waste materials, particularly plastic waste, increase the stiffness modulus and thus the strength of the pavement. However, researchers have also found that the stiffness modulus decreases above a certain ratio of waste material to virgin material [25–28]. The Slovenian technical specifications for the design of new asphalt pavements, based on the empirical AASHTO method, contain

a series of diagrams showing a spectrum of values for Marshall test results of asphalt mixtures and the CBR values of base and sub-base mixtures [29]. These diagrams were considered in the analyses in this paper (see Figure 2). The Marshall test diagram and a range of CBR values for the base and sub-base are paired with equivalence factors that affect pavement design.

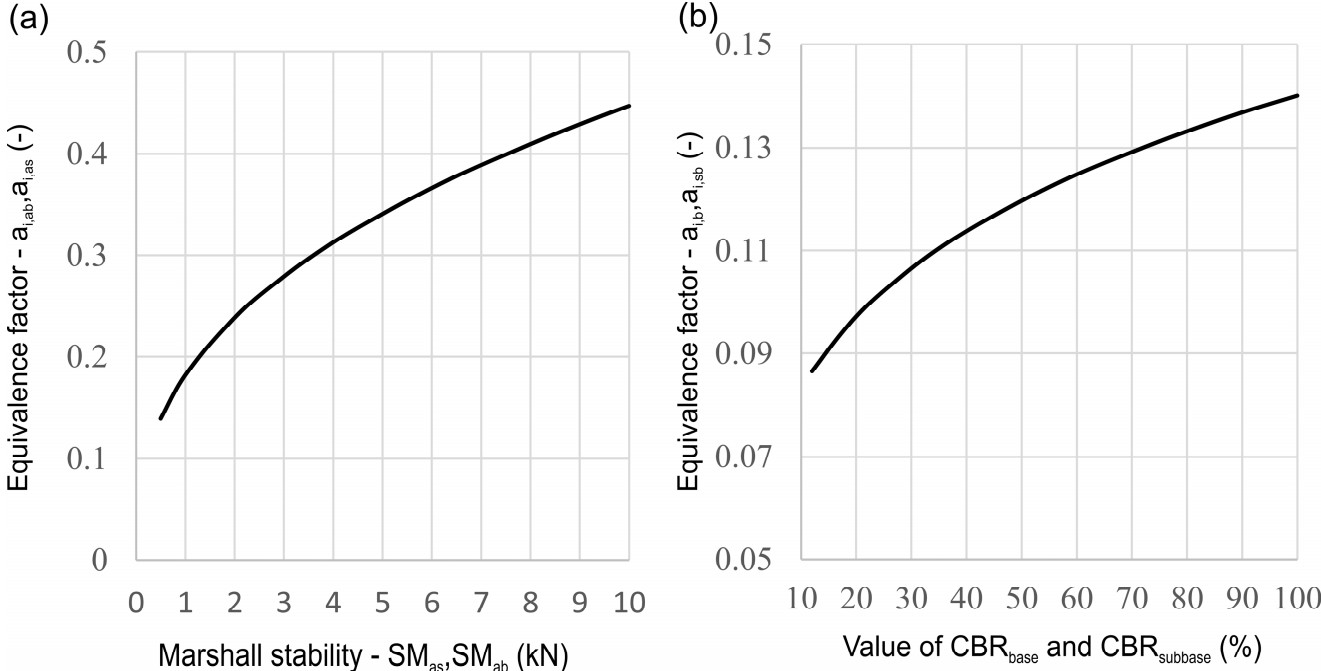

**Figure 2.** Equivalence factors for given values of (**a**) Marshall stability of asphalt and (**b**) CBR of base and sub-base material.

To show the Marshall stability for asphalt pavements with different waste materials, a diagram (Figure 3) was prepared based on several authors. The diagram includes the ratio of waste materials to virgin materials and the effects on Marshall stability. Fly ash was analyzed and showed an increase of 4 kN in the Marshall stability up to a certain point of the waste ratio when mixed with bitumen, but the addition of additional fly ash, above 7% of the mixture, was found to be suboptimal [30]. Plastic waste in the form of PE carry bags also showed a similar effect on the Marshall stability, but the inflection point was 12% of the bitumen weight, indicating a larger amount of waste that can be added to the asphalt pavement and make it more environmentally friendly [31]. To further study plastic waste, PET bottles were used in combination with bitumen. The test showed a steady increase in Marshall stability with the turning point being at around 12% of PET bottles added [32]. The results are consistent with the previous study on PE carry bags, thus proving that plastic waste is a reliable way to replace virgin material in asphalt pavements. Reinforcing asphalt mixtures with Electric Arc Furnace Dust (EAFD), a hazardous waste generated in the metallurgical industry, proved to be efficient for reinforcing asphalt pavements and could be a cost-effective and sustainable way to reduce the potential environmental impact of this waste material [33].

The inclusion of waste material in base and sub-base layers is rarely discussed, and how mixing waste and gravel in various proportions affects CBR values is rarely stated. The paper considers the experience of the authors of previous studies. An analysis of permanent deformations of the base course of pavements and their dependence on the proportion of accessors in the stone mixture had been carried out [34]. Žlender and Trauner [35] presented the use of the electro-filter granular (EFG) for the base course of pavements. Different types of EFG specimens were studied, first without reinforcement and then with one, three, and five reinforcements. The failure and deformation envelope curves are given separately for

all types of EFG specimens and compared with the results performed for standard ballast base courses. The use of geosynthetic geocells as base course reinforcement was studied. The results show that the proper use and positioning of geocells filled with supplementary material can significantly reduce the thickness of asphalt layers [36,37]. The Cinderella project [38] aimed to develop a new Circular Economy Business Model for the use of secondary raw materials in urban areas. It included the use of secondary raw materials created by recycling construction, industrial, mining, and some municipal wastes. One of the project objectives was to create a circular model for resource use that would reduce negative impacts on the environment. This would be achieved by introducing circular supply chains in urban construction. The URGE project [39] accelerates the transition to a circular economy. It aims to develop integrated urban circular economy strategies in the construction sector as the main consumer of raw materials. Research on lightweight materials as a possible solution to improve the low-bearing-capacity subgrade of pavements has concluded that, due to their low density, lightweight materials can be used to reduce weight in areas with the low-bearing-capacity subgrade. Due to their high porosity and the resulting good thermal insulation properties, lightweight materials can be used as frost protection layers in cold regions or as thermal insulation layers under road pavements. From a mechanical point of view, this material can provide good compressive strength and stiffness due to the relatively thick pore walls compared to the diameter of the voids [40].

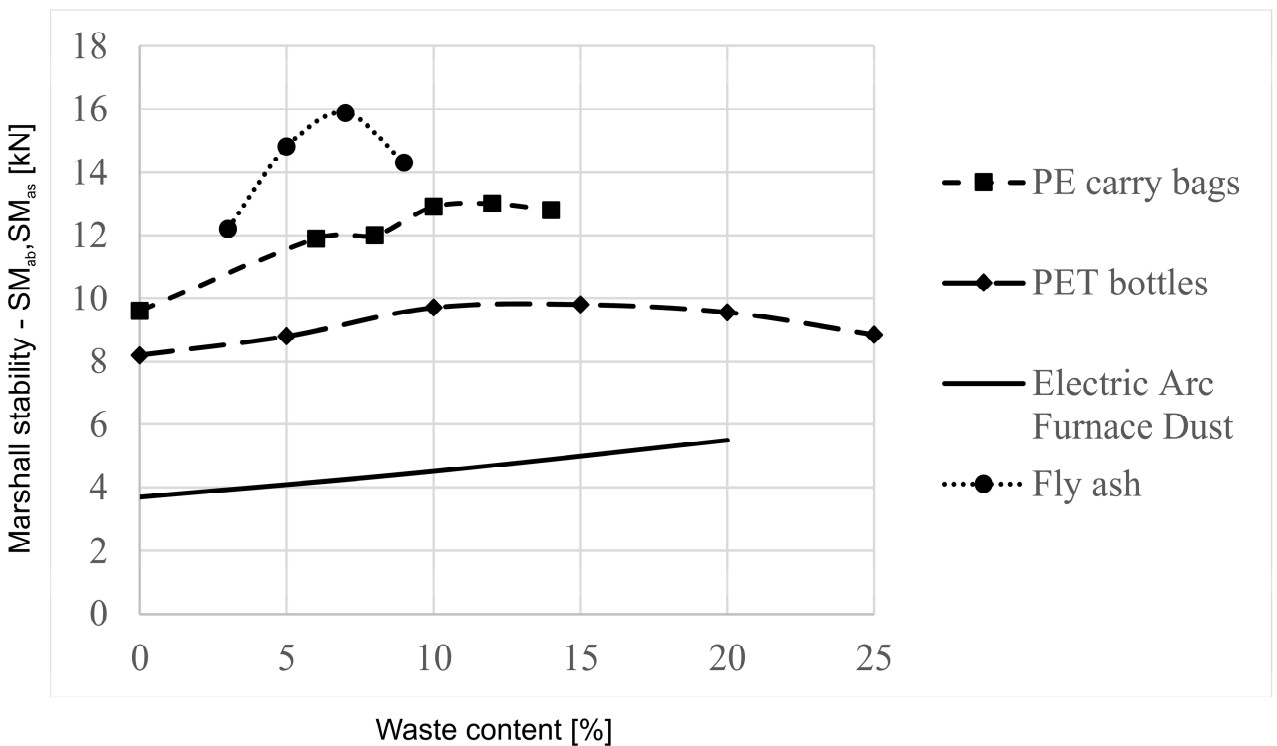

**Figure 3.** Various test results of SM for different waste materials and their content in the bituminous mix.

## 3. Optimization Model of Pavement Structure

The design and construction of pavement structures are of great importance due to several interrelated factors. First, it is important to create a pavement structure that can withstand the expected traffic loads during its lifespan. To achieve this, the criteria set forth in the design specifications must be followed. By carefully considering these criteria, engineers can ensure that the pavement structure will withstand the expected traffic loads and maintain its structural integrity. Furthermore, another significant objective is to construct the pavement structure while minimizing construction costs. To achieve this, a cost objective function is defined for the pavement structure, which can be minimized using optimization algorithms. By using such algorithms, engineers can determine the

most cost-effective pavement design and materials, thereby reducing the financial burden associated with pavement construction.

Moreover, it is also crucial to address environmental concerns during the pavement construction process. Alongside minimizing construction costs, efforts should be made to decrease the carbon footprint of the pavement structure. This can be achieved by implementing sustainable practices, utilizing environmentally friendly materials, and optimizing construction processes to reduce energy consumption and emissions. In addition, it is critical that environmental considerations be taken into account when constructing pavements. In addition to minimizing construction costs, efforts should be made to reduce the carbon footprint of pavement construction. This can be achieved by adopting sustainable practices, minimizing pavement thickness, using environmentally friendly materials, and optimizing construction processes to reduce energy consumption and emissions. To effectively achieve the aforementioned objectives, the development of an optimization model is required. This model should include a cost objective function that considers construction costs and a carbon footprint function that considers environmental impacts. In addition, the optimization model should include various constraints that limit the objective function to ensure that the design of a pavement structure meets the required standards and specifications. By incorporating these elements into the optimization model, engineers can aim for a pavement structure that is both structurally sound and environmentally sustainable while minimizing construction costs.

Such an optimization model is shown in Table 1, where the cost objective function of the pavement design is developed, and the $CO_2$ emissions generated during construction are evaluated. The optimized pavement structure meets all required standards and specifications based on an empirical pavement design method.

**Table 1.** Optimization model.

| | |
|---|---|
| Objective function | $COST_{pav} = C_{exc} + C_{gc} + C_{fill,b} + C_{as,subs} + C_{fill,sb} + C_{as} + C_{ab} + C_{geo}$ <br> $C_{exc} = c_{exc} \cdot h_{total} \cdot (B_{ve} + B_{as}) \cdot L$ <br> $C_{gc} = c_{gc} \cdot (B_{ve} + B_{as}) \cdot L$ <br> $C_{fill,b} = c_{fill,b} \cdot (B_{ve} + B_{as}) \cdot d_b \cdot L$ <br> $C_{as,subs} = c_{as,subs} \cdot B_{as} \cdot L$ <br> $C_{fill,sb} = c_{fill,sb} \cdot (B_{ve} + B_{as}) \cdot d_{sb} \cdot L$ <br> $C_{as} = c_{as} \cdot B_{as} \cdot d_{as} \cdot L$ <br> $C_{ab} = c_{ab} \cdot B_{as} \cdot d_{ab} \cdot L$ <br> $C_{geo} = c_{geo} \cdot (B_{ve} + B_{as}) \cdot L$ |
| $CO_2$ emissions | $CO_{2,\,total} = CO_{2,exc} + CO_{2,fill,b} + CO_{2,as,subs} + CO_{2,fill,sb} + CO_{2,as} + CO_{2,ab} + CO_{2,geo}$ <br> $CO_{2,exc} = ci_{exc} \cdot h_{total} \cdot (B_{ve} + B_{as}) \cdot L$ <br> $CO_{2,fill,b} = ci_{fill,b} \cdot (B_{ve} + B_{as}) \cdot d_b \cdot L \cdot \rho_{base}$ <br> $CO_{2,\,as,subs} = ci_{as,subs} \cdot B_{as} \cdot L$ <br> $CO_{2,fill,sb} = ci_{fill,sb} \cdot (B_{ve} + B_{as}) \cdot d_{sb} \cdot L \cdot \rho_{sub-base}$ <br> $CO_{2,as} = ci_{as} \cdot B_{as} \cdot d_{as} \cdot L \cdot \rho_{as}$ <br> $CO_{2,ab} = ci_{ab} \cdot B_{as} \cdot d_{ab} \cdot L \cdot \rho_{ab}$ <br> $CO_{2,geo} = ci_{geo} \cdot (B_{ve} + B_{as}) \cdot L$ |
| Condition 1 | $D_{total} \geq D_{req}$ <br> $D_{total} = d_{as} \cdot a_{i,as} + d_{ab} \cdot a_{i,ab} + d_b \cdot a_{i,b}$ <br> $D_{req} = d_{asb,0} \cdot 0.38 + d_{b,CBR_{mod}} \cdot 0.14$ <br> $d_{asb,0} = a_1 \cdot T_n{}^{a_2}$ <br> $d_{b,CBR_{mod}} = \left( (c_1 - c_2 \cdot CBR_{mod}) \cdot \ln(T_n) - c_3 + e^{(c_4 \cdot CBR_{mod}) \cdot c_5} \right) / \gamma_{geo,b}$ |
| Condition 2 | $D_{total,AC} \geq D_{req,AC}$ <br> $D_{total,AC} = d_{as} \cdot a_{i,as} + d_{ab} \cdot a_{i,ab}$ <br> $D_{req,AC} = d_{asb,0} \cdot 0.38$ |

**Table 1.** *Cont.*

| Condition 3 | $D_{total,base} \geq D_{req,base}$<br>$D_{total,base} = d_b \cdot a_{i,b}$<br>$D_{req,base} = d_{b,CBR_{mod}} \cdot 0.14$ |
|---|---|
| Condition 4 | $d_{prov} \geq d_{asb,0}$<br>$d_{prov} = d_{as} + d_{ab}$ |
| Condition 5 | $d_b \geq d_{b,req}$<br>$d_{b,req} = d_{b,CBR_{mod}}$ |
| Condition 6 | $h_{total} \geq h_{req}$<br>$h_{total} = d_{as} + d_{ab} + d_b + d_{sb}$<br>$h_{req} = h_m \cdot f_{fr}$ |
| Condition 7 | $d_{sb} \geq d_{sb,CBR_{mod}}$<br>$d_{sb,CBR_{mod}} = \left( b_1 \cdot \left( \frac{b_2 \cdot (CBR_{mod} - CBR)}{b_3 - CBR} + b_4 \right) \cdot \left( \frac{0.14}{a_{i,sb}} \right) \right) / \gamma_{geo,sb}$ |
| Condition 8 | $d_{as} \geq d_{as,min}$ |
| Condition 9 | $d_{ab} \geq d_{ab,min}$ |
| Condition 10 | $d_b \geq d_{b,min}$ |
| Condition 11 | $d_{sb} \geq d_{sb,min}$ |

Figure 4 shows the flexible pavement structure, which contains two bound layers (asphalt surface layer and asphalt binder layer) and two unbound layers (base course and sub-base course), as well as geosynthetic reinforcement.

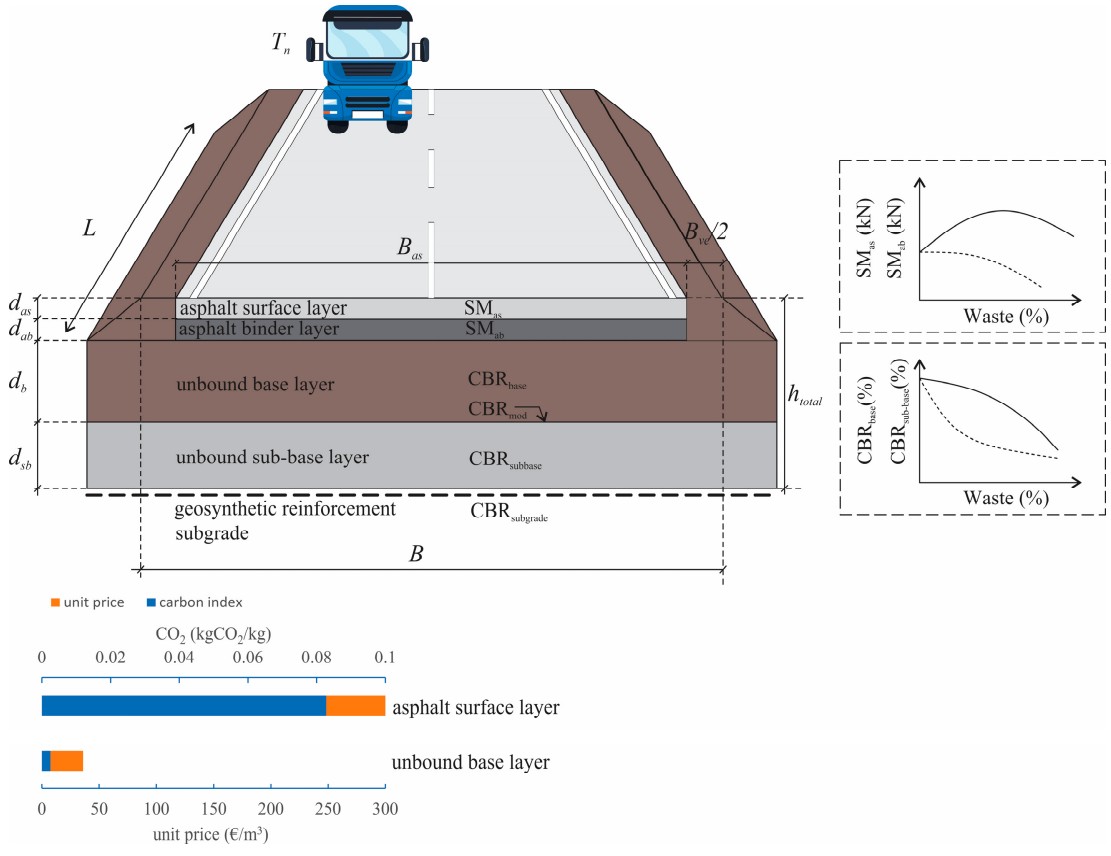

**Figure 4.** Cross-section of the road pavement structure.

The optimization model comprises four variables that correspond to distinct layers within a pavement structure. The variable $d_{as}$ (m) represents the asphalt surface layer, $d_{ab}$ (m) represents the binder layer of the asphalt, $d_b$ (m) represents the unbound base layer, and $d_{sb}$ (m) represents the unbound sub-base layer. Additionally, the construction cost of the pavement structure is defined as $COST_{pav}$ (EUR). The input data provided are used to represent various parameters and characteristics relevant to the design properties, which are used in the optimization process of the pavement structure. The definitions of the individual input data and the associated values are listed in Table 2. The unit prices ($c_{exe}$, $c_{gc}$, $c_{fill,sb}$, $c_{fill,b}$, $c_{as,subs}$, $c_{as}$, $c_{ab}$, and $c_{geo}$) of each material used in the pavement structure were assigned, as well as carbon indices ($ci_{exe}$, $ci_{fill,sb}$, $ci_{fill,b}$, $ci_{as,subs}$, $ci_{as}$, $ci_{ab}$, and $ci_{geo}$) that allow the calculation of the total $CO_2$ emissions of the materials used in the pavement structure. While the empirical pavement design method is included in the optimization model, where the pavement thickness is limited by the pavement thickness index, it is of utmost importance to determine the equivalence factors $a_{i,as}$, $a_{i,ab}$, $a_{i,b}$, and $a_{i,sb}$ based on the material properties of the individual pavement layers. In this way, it was possible to take into account the inclusion of waste material affecting the stability according to the Marshall stability test ($SM_{as}$ and $SM_{ab}$) in the asphalt concrete layer and the CBR value ($CBR_{base}$ and $CBR_{subbase}$) for the unbound base and the sub-base layer.

$$a_{i,as} = t_1 \cdot SM_{as}{}^{t_2} \tag{1}$$

$$a_{i,ab} = t_3 \cdot SM_{ab}{}^{t_4} \tag{2}$$

$$a_{i,b} = t_5 \cdot CBR_{base}{}^{t_6} \tag{3}$$

$$a_{i,sb} = t_7 \cdot CBR_{subbase}{}^{t_8} \tag{4}$$

To satisfy the first five conditions, the required thickness of the asphalt, $d_{asb,0}$ (cm), and the required thickness of the unbound base layer, $d_{b,CBR,mod}$ (cm), which depend on the CBR of the subgrade, must be determined by a parameterized function. Therefore, the parameters $a_1$, $a_2$, $c_1$, $c_2$, $c_3$, $c_4$, and $c_5$ were determined based on an approximation of the charts included in the technical specifications for roads, which relate to the required thickness and the number of ESALs $T_n$. Since the main objective of placing the sub-base layer is to improve the CBR value of the subgrade, the thickness of the required unbound sub-base layer $d_{sb,CBR,mod}$ (cm) is calculated to provide the modified CBR value at the top of the sub-base layer $CBR_{mod}$. The correlation between the original CBR value of the subgrade, the thickness of the sub-base, and the modified CBR value at the top of the sub-base is determined by the parameters $b_1$, $b_2$, $b_3$, and $b_4$. Furthermore, the thickness of the sub-base layer can be reduced by a factor of $\gamma_{geo,sb}$ if the geosynthetic reinforcement is installed in the contact between the subgrade and the sub-base layer. Condition 6 ensures that the overall thickness of the pavement is sufficient to be frost resistant. While frost depth depends on geographic location and hydrologic conditions, factors $f_{fr}$ and $h_m$ are determined using Table 3. Hydrological conditions are favorable if the total thickness of the pavement structure is at least 1.5 m, the water table is constantly below freezing, and drainage is ensured without water inflow within the pavement. Otherwise, the factors for unfavorable conditions must be considered. The last four conditions (conditions 8–11) ensure that the thickness of each pavement layer is of a sufficient minimum thickness according to conventional pavement construction techniques.

**Table 2.** Explanation of input data in optimization model along with determined values.

| Symbol | Value | Description |
|---|---|---|
| $c_{exe}$ (€/m$^3$) | 9 | unit price of the ground excavation |
| $c_{gc}$ (€/m$^2$) | 2.5 | unit price of the ground compaction |
| $c_{fill,sb}$ (€/m$^3$) | 24 | unit price of the unbound sub-base fill |
| $c_{fill,b}$ (€/m$^3$) | 36 | unit price for unbound base fill |
| $c_{as,subs}$ (€/m$^2$) | 1.5 | unit price of the asphalt substrate |
| $c_{as}$ (€/m$^3$) | 300 | unit price of the asphalt surface layer |
| $c_{ab}$ (€/m$^3$) | 200 | unit price of the asphalt binder layer |
| $c_{geo}$ (€/m$^2$) | 3.2 | unit price of the geosynthetics |
| $B_{ve}$ (m) | 1 | width of the verge |
| $B_{as}$ (m) | 8 | width of the asphalt surface |
| $L$ (m) | 1000 | length of pavement sections |
| $ci_{exe}$ (kgCO$_2$/m$^3$) | 1.38 | carbon index for the ground excavation |
| $ci_{fill,b}$ (kgCO$_2$/kg) | 0.00248 | carbon index for unbound sub-base fill |
| $ci_{as,subs}$ (kgCO$_2$/m$^2$) | 0.35 | carbon index for unbound base fill |
| $ci_{fill,sb}$ (kgCO$_2$/kg) | 0.00248 | carbon index for asphalt substrate |
| $ci_{as}$ (kgCO$_2$/kg) | 0.08278 | carbon index for asphalt surface layer |
| $ci_{ab}$ (kgCO$_2$/kg) | 0.08278 | carbon index for asphalt binder layer |
| $ci_{geo}$ (kgCO$_2$/m$^2$) | 0.396 | carbon index for geosynthetics |
| $\rho_{base}$ (kg/m$^3$) | 1800 | density of the unbound base fill |
| $\rho_{sub-base}$ (kg/m$^3$) | 1800 | density of the unbound sub-base fill |
| $\rho_{as}$ (kg/m$^3$) | 2400 | density of the asphalt surface layer |
| $\rho_{ab}$ (kg/m$^3$) | 2400 | density of the asphalt binder layer |
| $t_1 = t_3$ (-) | 0.182104767 | parameter for $a_{i,as}$ and $a_{i,ab}$ determination |
| $t_2 = t_4$ (-) | 0.389702035 | parameter for $a_{i,as}$ and $a_{i,ab}$ determination |
| $t_5 = t_7$ (-) | 0.049219606 | parameter for $a_{i,b}$ and $a_{i,sb}$ determination |
| $t_6 = t_8$ (-) | 0.227144669 | parameter for $a_{i,b}$ and $a_{i,sb}$ determination |
| $a_1$ (-) | 0.6567 | parameter for required thickness of the asphalt |
| $a_2$ (-) | 0.2175 | parameter for required thickness of the asphalt |
| $b_1$ (-) | 8.382 | parameter for required thickness of sub-base |
| $b_2$ (-) | −0.791 | parameter for required thickness of sub-base |
| $b_3$ (-) | 1.975 | parameter for required thickness of sub-base |
| $b_4$ (-) | 1.912 | parameter for required thickness of sub-base |
| $c_1$ (-) | 6.239 | parameter for required thickness of base layer |
| $c_2$ (-) | 0.376 | parameter for required thickness of base layer |
| $c_3$ (-) | 26.64 | parameter for required thickness of base layer |
| $c_4$ (-) | 0.141 | parameter for required thickness of base layer |
| $c_5$ (-) | 4.882 | parameter for required thickness of base layer |
| $CBR_{mod}$ (%) | 15.0 | modified CBR value at the top of the sub-base layer |
| $\gamma_{geo,sb}$ (-) | 2.0 | reduction factor for the consideration of the geosynthetic |
| $h_m$ (cm) | 80 | depth of frost penetration |
| $f_{fr}$ (-) | 0.8 | Factor for the conditions of the material at the site |
| $d_{as,min}$ (m) | 0.04 | minimum thickness of the asphalt surface layer |
| $d_{ab,min}$ (m) | 0.06 | minimum thickness of the asphalt binder layer |
| $d_{b,min}$ (m) | 0.25 | minimum thickness of the unbound base layer |
| $d_{sb,min}$ (m) | 0.20 | minimum thickness of the unbound sub-base layer |

**Table 3.** Minimum required thicknesses of the road pavement structures $h_{req}$ [29].

| Resistance of the Material under the Pavement Structure against the Effects of Freezing and Thawing | Hydrological Conditions | Minimum Thickness of Pavement Structure $h_{req} = (f_{fr}) \cdot h_m$ $h_m$ Is Depth of Frost Penetration | |
|---|---|---|---|
| | | **to an Altitude of 600 m** | **from an Altitude of 600 m** |
| resistant | favorable | $(0.6) \cdot h_m$ | $(0.7) \cdot h_m$ |
| | unfavorable | $(0.7) \cdot h_m$ | $(0.8) \cdot h_m$ |
| not resistant | favorable | $(0.7) \cdot h_m$ | $(0.8) \cdot h_m$ |
| | unfavorable | $(0.8) \cdot h_m$ | $(0.9) \cdot h_m$ |

## 4. Multi Parametric Optimization

The main objective of this work is to analyze how the material properties of each layer of the pavement affect the design thickness of the pavement and consequently the construction costs and $CO_2$ emissions. An optimization model was used to determine the minimum thickness of each layer that still meets all conditions and consequently ensures sufficient performance over the intended 20-year period. The ESAL is the most important design parameter in pavement design, so a parametric analysis was also performed for this parameter. Therefore, the optimal designs of the pavement structure were determined for 450 combinations of the following parameters:

- Total number of ESALs: $T_n$ ($1 \times 10^4$; $1 \times 10^5$; $1 \times 10^6$; $1 \times 10^7$; $1 \times 10^7$).
- California Bearing Ratio of subgrade: *CBR* (3%; 4%; 5%; 6%; 7%).
- Marshall stability of asphalt layers: $SM_{as} = SM_{ab}$ (2 kN; 4 kN; 6 kN; 8 kN; 10 kN).
- California Bearing Ratio of unbound layers: $CBR_{base} = CBR_{subbase}$ (100%; 60%; 30%).

By performing such a multiparametric analysis, it is possible to show how the thickness of each pavement layer increases as the material properties decrease due to the incorporation of waste material into each layer of the pavement structure under different traffic loads. The optimization model was developed so that different values of Marshall stability could be applied to the asphalt surface layer and the asphalt binder layer. However, in the parametric analysis, the Marshall stability takes the same value for the asphalt surface and asphalt binder layer ($SM_{as} = SM_{ab}$). The same applies to the unbound base and the sub-base layer ($CBR_{base} = CBR_{subbase}$).

The results of the optimal solution are presented in several steps. First, the parallel coordinate plot is used to present multidimensional data on Marshall stability, CBR of unbound layers, pavement cost, and $CO_2$ emissions. In a parallel coordinate plot, the data points are represented as contiguous lines, and the parallel axes represent the different variables (Marshall stability, CBR of unbound layers, values of optimal pavement cost, and $CO_2$ emissions).

The data points are grouped based on Marshall stability and plotted in different colors. Figure 5a shows fifteen different optimal costs and $CO_2$ emissions for different combinations of Marshall stability and CBR of the unbound layers for an ESAL of $T_n = 1 \times 10^6$ and for subgrade $CBR_{subgrade} = 3\%$. While Figure 5b shows the results for pavements with geosynthetics, Figure 5a presents the results without geosynthetic reinforcement. The parallel plot can be read as in Figure 5a (see the blue lines) as follows: For a Marshall stability of SM = 2 kN and a $CBR_{base} = 30\%$, the optimal construction cost is 95.5 €/m$^2$ and $CO_2$ emissions are 41 kgCO$_2$/m$^2$, while for the same SM = 2 kN and a better $CBR_{base} = 100\%$, the optimal construction cost is 85 €/m$^2$ and $CO_2$ emissions are 39.3 kgCO$_2$/m$^2$. In this case, reducing the CBR of the unbound layer from 100% to 30% means an increase of 12% in costs and 4% in $CO_2$ emissions. Similarly, for a pavement structure with geosynthetic reinforcement, and the reduction in the quality of the unbound layers increases the cost by 9% and the $CO_2$ emissions by 3%. It was also found that Marshall stability has the largest impact on both cost and $CO_2$ emissions.

From the results shown in Figure 6, it can be seen that costs increase by 25% and $CO_2$ emissions by 48% when the Marshal stability of the asphalt layer is reduced from 10 kN to 2 kN. This reduction was calculated for a pavement with geosynthetics and a CBR value of 30% for unbound layers. The use of geosynthetics in most of the cases discussed in the parametric analysis reduces the cost of pavement structure and the amount of $CO_2$ emissions. The largest reduction in COST and $CO_2$ is given in the case where $T_n$ is in the range of $1 \times 10^4$, CBR = 3%, SM = 10 kN, and the CBR of the base and sub-base layers is 30%. In this case, the use of geosynthetics results in a 15% reduction in COST and a 9% reduction in $CO_2$ due to the reduced thickness of the unbound pavement structure. Figure 6 shows that the use of geosynthetics is economically justified when the CBR of the subgrade is less than 5%, and that the use of geosynthetics is environmentally justified when the $CBR_{subgrade}$ is less than 6% if the properties of the base and sub-base layer are assumed to be $CBR_{base,subbase} = 100\%$. If the CBR value of the base and sub-base layer

is low ($CBR_{base,subbase}$ = 30%), the use of geosynthetics is justified from an economic and environmental point of view if the CBR value of subgrade is less than 7%. It was found that the use of geosynthetics is particularly important in the case where the base and subgrade layers partially contain waste materials.

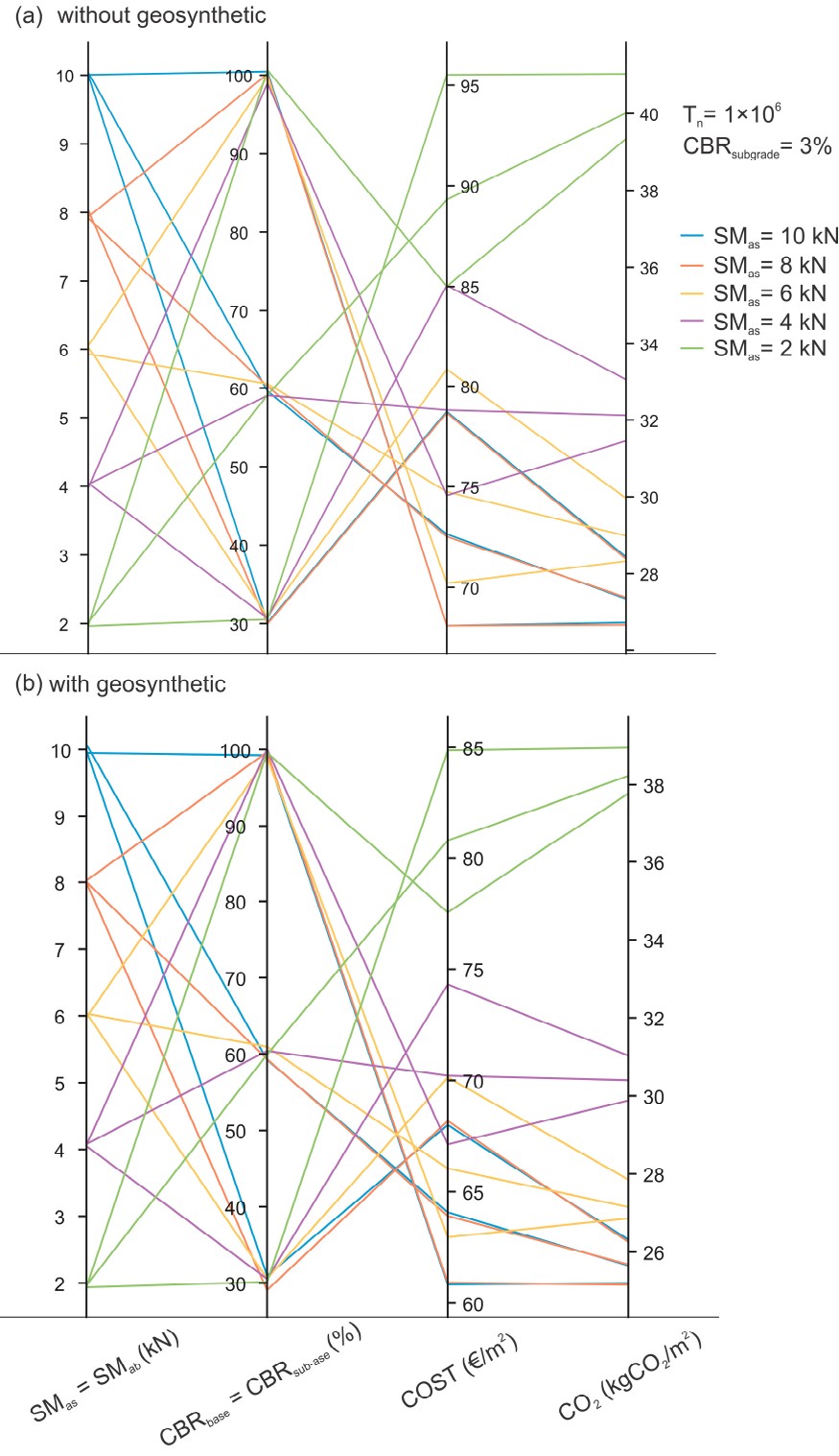

**Figure 5.** Parallel coordinate plot of optimal pavement cost and $CO_2$ emissions for different Marshall stability of asphalt layer and CBR of unbound layers: (**a**) without geosynthetic reinforcement and (**b**) with geosynthetic reinforcement.

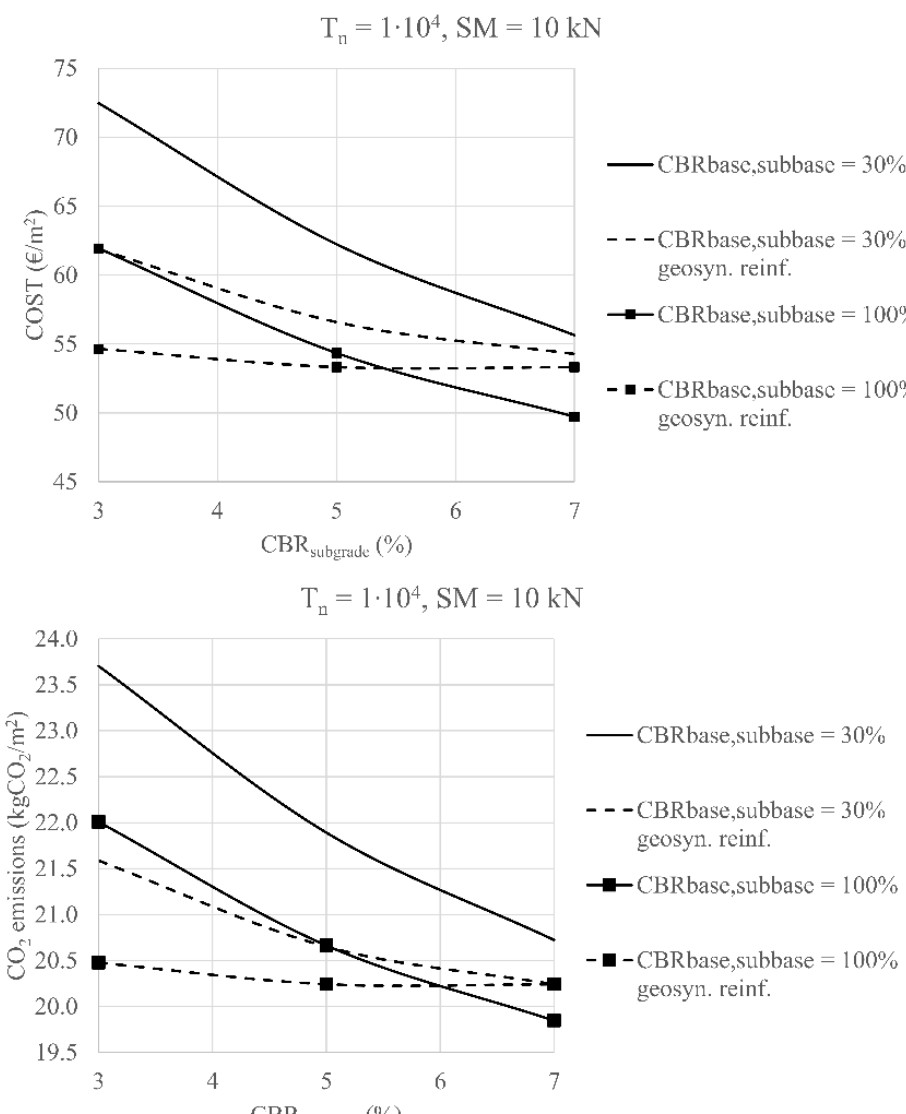

**Figure 6.** Costs and $CO_2$ emissions of road pavements depend on the quality of the subgrade and the use of geosynthetics.

Table 4 (without geosynthetics) and Table 5 (with geosynthetics) show the optimal pavement design including the thickness of asphalt layers, the base layer, and the sub-base, as well as pavement costs and $CO_2$ emissions. It should be noted that for all parameter combinations, the optimal thickness of the asphalt surface layer was calculated as $d_{as} = 4$ cm, which corresponds to the minimum value specified. Based on these two tables, it was possible to evaluate the effects of the Marshall stability and the CBR value of the unbound layer on the design of flexible pavements. The thickness of the unbound layers increased from 91 cm to 120 cm when the CBR value of the base and sub-base layers decreased from 100% to 30%. For real-world pavement projects, Tables 4 and 5 can help engineers and designers select the most appropriate materials, layer thicknesses, and construction methods for pavements where the design is based on the minimum cost and $CO_2$ emissions. The model was developed in a general form that allows an optimal design to be obtained for any input data based on real site conditions, material properties, and traffic loads.

**Table 4.** Optimal pavement design for subgrade $CBR_{subgrade} = 3\%$, without geosynthetic.

| | | $CBR_{base} = CBR_{subbase} = 100\%$ | | | | | $CBR_{base} = CBR_{subbase} = 30\%$ | | | | |
|---|---|---|---|---|---|---|---|---|---|---|---|
| $T_n$ (ESAL) | SM | $d_{as} + d_{ab}$ | $d_b$ | $d_{sb}$ | COST | $CO_2$ | $d_{as} + d_{ab}$ | $d_b$ | $d_{sb}$ | COST | $CO_2$ |
| (-) | (kN) | (cm) | (cm) | (cm) | (€/m²) | (kgCO₂/m²) | (cm) | (cm) | (cm) | (€/m²) | (kgCO₂/m²) |
| $1.0 \times 10^4$ | 10 | 10 | 25 | 66 | 61.9 | 22.0 | 10 | 33 | 87 | 72.5 | 23.7 |
| $1.0 \times 10^5$ | 10 | 10 | 25 | 66 | 61.9 | 22.0 | 10 | 33 | 87 | 72.5 | 23.7 |
| $1.0 \times 10^6$ | 10 | 13 | 25 | 66 | 68.2 | 26.7 | 13 | 33 | 87 | 78.7 | 28.4 |
| $1.0 \times 10^7$ | 10 | 22 | 25 | 66 | 87.0 | 40.9 | 22 | 33 | 87 | 97.5 | 42.6 |
| $1.0 \times 10^8$ | 10 | 36 | 25 | 66 | 116.3 | 62.9 | 36 | 33 | 87 | 126.8 | 64.6 |
| $1.0 \times 10^4$ | 8 | 10 | 25 | 66 | 61.9 | 22.0 | 10 | 33 | 87 | 72.5 | 23.7 |
| $1.0 \times 10^5$ | 8 | 10 | 25 | 66 | 61.9 | 22.0 | 10 | 33 | 87 | 72.5 | 23.7 |
| $1.0 \times 10^6$ | 8 | 13 | 25 | 66 | 68.2 | 26.7 | 13 | 33 | 87 | 78.7 | 28.4 |
| $1.0 \times 10^7$ | 8 | 22 | 25 | 66 | 87.0 | 40.9 | 22 | 33 | 87 | 97.5 | 42.6 |
| $1.0 \times 10^8$ | 8 | 36 | 25 | 66 | 116.3 | 62.9 | 36 | 33 | 87 | 126.8 | 64.6 |
| $1.0 \times 10^4$ | 6 | 10 | 25 | 66 | 61.9 | 22.0 | 10 | 33 | 87 | 72.5 | 23.7 |
| $1.0 \times 10^5$ | 6 | 10 | 25 | 66 | 61.9 | 22.0 | 10 | 33 | 87 | 72.5 | 23.7 |
| $1.0 \times 10^6$ | 6 | 14 | 25 | 66 | 70.3 | 28.3 | 14 | 33 | 87 | 80.8 | 30.0 |
| $1.0 \times 10^7$ | 6 | 23 | 25 | 66 | 89.1 | 42.4 | 23 | 33 | 87 | 99.6 | 44.1 |
| $1.0 \times 10^8$ | 6 | 38 | 25 | 66 | 120.5 | 66.0 | 38 | 33 | 87 | 131.0 | 67.7 |
| $1.0 \times 10^4$ | 4 | 10 | 25 | 66 | 61.9 | 22.0 | 10 | 33 | 87 | 72.5 | 23.7 |
| $1.0 \times 10^5$ | 4 | 10 | 25 | 66 | 61.9 | 22.0 | 10 | 33 | 87 | 72.5 | 23.7 |
| $1.0 \times 10^6$ | 4 | 16 | 25 | 66 | 74.5 | 31.4 | 16 | 33 | 87 | 85.0 | 33.1 |
| $1.0 \times 10^7$ | 4 | 27 | 25 | 66 | 97.5 | 48.7 | 27 | 33 | 87 | 108.0 | 50.4 |
| $1.0 \times 10^8$ | 4 | 44 | 25 | 66 | 133.0 | 75.4 | 44 | 33 | 87 | 143.5 | 77.1 |
| $1.0 \times 10^4$ | 2 | 13 | 25 | 66 | 68.2 | 26.7 | 13 | 33 | 87 | 78.7 | 28.4 |
| $1.0 \times 10^5$ | 2 | 13 | 25 | 66 | 68.2 | 26.7 | 13 | 33 | 87 | 78.7 | 28.4 |
| $1.0 \times 10^6$ | 2 | 21 | 25 | 66 | 84.9 | 39.3 | 21 | 33 | 87 | 95.5 | 41.0 |
| $1.0 \times 10^7$ | 2 | 36 | 25 | 66 | 116.3 | 62.9 | 36 | 33 | 87 | 126.8 | 64.6 |
| $1.0 \times 10^8$ | 2 | 58 | 25 | 66 | 162.3 | 97.4 | 58 | 33 | 87 | 172.8 | 99.1 |

**Table 5.** Optimal geosynthetic reinforced pavement design for subgrade CBR = 3%.

| | | $CBR_{base} = CBR_{subbase} = 100\%$ | | | | | $CBR_{base} = CBR_{subbase} = 30\%$ | | | | |
|---|---|---|---|---|---|---|---|---|---|---|---|
| $T_n$ (ESAL) | SM | $d_{as} + d_{ab}$ | $d_b$ | $d_{sb}$ | COST | $CO_2$ | $d_{as} + d_{ab}$ | $d_b$ | $d_{sb}$ | COST | $CO_2$ |
| (-) | (kN) | (cm) | (cm) | (cm) | (€/m²) | (kgCO₂/m²) | (cm) | (cm) | (cm) | (€/m²) | (kgCO₂/m²) |
| $1.0 \times 10^4$ | 10 | 10 | 25 | 33 | 54.6 | 20.5 | 10 | 33 | 44 | 61.9 | 21.6 |
| $1.0 \times 10^5$ | 10 | 10 | 25 | 33 | 54.6 | 20.5 | 10 | 33 | 44 | 61.9 | 21.6 |
| $1.0 \times 10^6$ | 10 | 13 | 25 | 33 | 60.9 | 25.2 | 13 | 33 | 44 | 68.1 | 26.3 |
| $1.0 \times 10^7$ | 10 | 22 | 25 | 33 | 79.7 | 39.3 | 22 | 33 | 44 | 87.0 | 40.4 |
| $1.0 \times 10^8$ | 10 | 36 | 25 | 33 | 109.0 | 61.3 | 36 | 33 | 44 | 116.2 | 62.4 |
| $1.0 \times 10^4$ | 8 | 10 | 25 | 33 | 54.6 | 20.5 | 10 | 33 | 44 | 61.9 | 21.6 |
| $1.0 \times 10^5$ | 8 | 10 | 25 | 33 | 54.6 | 20.5 | 10 | 33 | 44 | 61.9 | 21.6 |
| $1.0 \times 10^6$ | 8 | 13 | 25 | 33 | 60.9 | 25.2 | 13 | 33 | 44 | 68.1 | 26.3 |
| $1.0 \times 10^7$ | 8 | 22 | 25 | 33 | 79.7 | 39.3 | 22 | 33 | 44 | 87.0 | 40.4 |
| $1.0 \times 10^8$ | 8 | 36 | 25 | 33 | 109.0 | 61.3 | 36 | 33 | 44 | 116.2 | 62.4 |
| $1.0 \times 10^4$ | 6 | 10 | 25 | 33 | 54.6 | 20.5 | 10 | 33 | 44 | 61.9 | 21.6 |
| $1.0 \times 10^5$ | 6 | 10 | 25 | 33 | 54.6 | 20.5 | 10 | 33 | 44 | 61.9 | 21.6 |
| $1.0 \times 10^6$ | 6 | 14 | 25 | 33 | 63.0 | 26.8 | 14 | 33 | 44 | 70.2 | 27.9 |
| $1.0 \times 10^7$ | 6 | 23 | 25 | 33 | 81.8 | 40.9 | 23 | 33 | 44 | 89.0 | 42.0 |
| $1.0 \times 10^8$ | 6 | 38 | 25 | 33 | 113.2 | 64.5 | 38 | 33 | 44 | 120.4 | 65.6 |
| $1.0 \times 10^4$ | 4 | 10 | 25 | 33 | 54.6 | 20.5 | 10 | 33 | 44 | 61.9 | 21.6 |
| $1.0 \times 10^5$ | 4 | 10 | 25 | 33 | 54.6 | 20.5 | 10 | 33 | 44 | 61.9 | 21.6 |
| $1.0 \times 10^6$ | 4 | 16 | 25 | 33 | 67.2 | 29.9 | 16 | 33 | 44 | 74.4 | 31.0 |
| $1.0 \times 10^7$ | 4 | 27 | 25 | 33 | 90.2 | 47.2 | 27 | 33 | 44 | 97.4 | 48.3 |
| $1.0 \times 10^8$ | 4 | 44 | 25 | 33 | 125.7 | 73.9 | 44 | 33 | 44 | 132.9 | 75.0 |
| $1.0 \times 10^4$ | 2 | 13 | 25 | 33 | 60.9 | 25.2 | 13 | 33 | 44 | 68.1 | 26.3 |
| $1.0 \times 10^5$ | 2 | 13 | 25 | 33 | 60.9 | 25.2 | 13 | 33 | 44 | 68.1 | 26.3 |
| $1.0 \times 10^6$ | 2 | 21 | 25 | 33 | 77.6 | 37.8 | 21 | 33 | 44 | 84.9 | 38.9 |
| $1.0 \times 10^7$ | 2 | 36 | 25 | 33 | 109.0 | 61.3 | 36 | 33 | 44 | 116.2 | 62.4 |
| $1.0 \times 10^8$ | 2 | 58 | 25 | 33 | 155.0 | 95.9 | 58 | 33 | 44 | 162.2 | 97.0 |

Different input data were utilized to determine the optimal configuration of the pavement structure. The four primary inputs comprise the overall count of ESAL ($T_n$), the Marshall stability of asphalt layers ($SM_{as} = SM_{ab}$), the California Bearing Ratio of the base and sub-base layer ($CBR_{base} = CBR_{subbase}$), and the subgrade conditions ($CBR_{subgrade}$). Through this multiparametric analysis, the primary goal was to employ these key attributes to anticipate other continuous characteristics, including the minimum cost of pavement structure (COST) and the minimum $CO_2$ emissions ($CO_2$). Prior to employing the predictive model, the dataset was split into a training dataset (odd-indexed samples) and a checking dataset (even-indexed samples). The "exhsrch" function in MATLAB (R2021a) was utilized to exhaustively search among the available inputs and determine the set of inputs that have the greatest impact on the optimal cost of the pavement structure and layer thickness. The "exhsrch" function involved building predictive models for each parameter combination, training them for an epoch, and subsequently reporting their achieved performance. In Figure 7, the leftmost input variable is the most pertinent in terms of the output, as it exhibits the lowest root-mean-square error (RMSE). The RMSE is defined as follows:

$$\text{RMSE} = \sqrt{\frac{\sum_{i=1}^{n}(\hat{x}_i - x_i)^2}{n}} \tag{5}$$

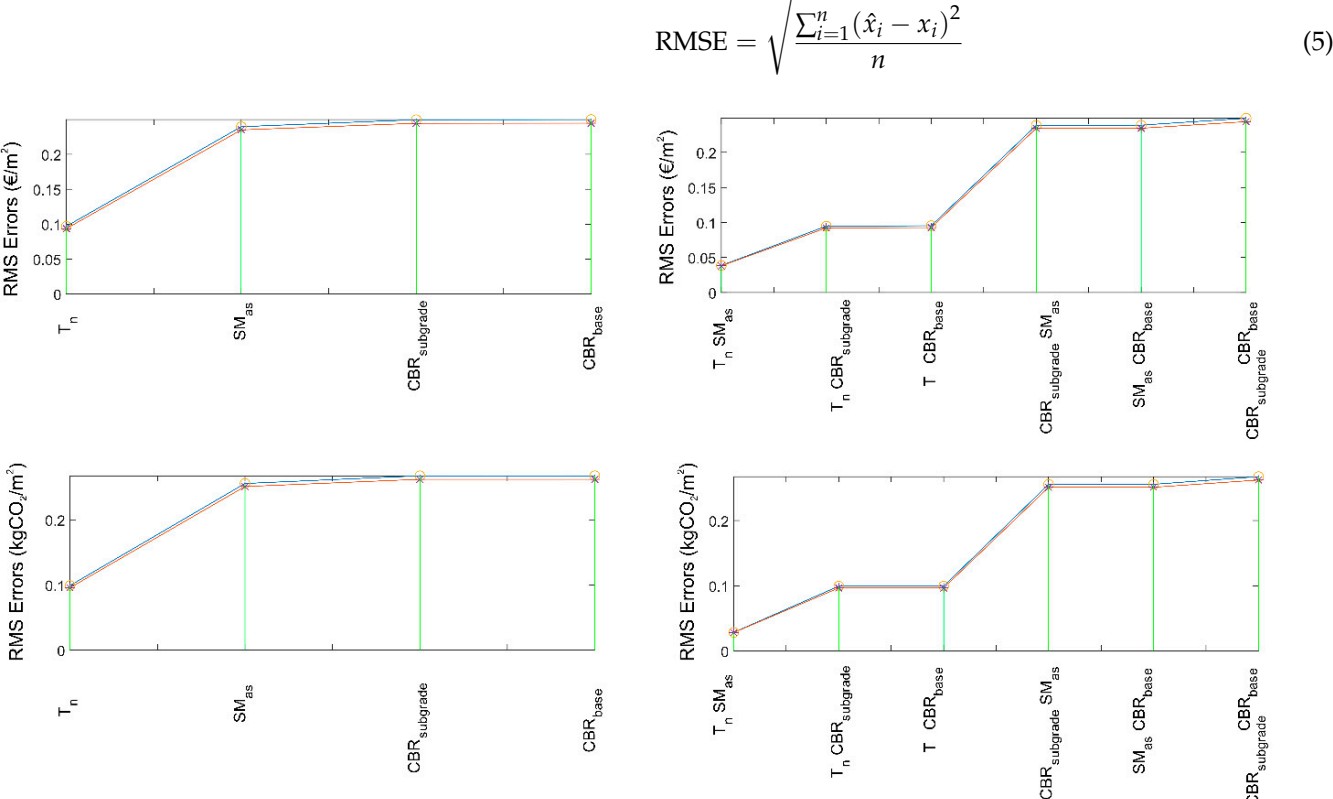

**Figure 7.** Influence of each input variable on the optimal cost of a pavement structure and the $CO_2$ emissions (training data in blue line, test data in red line).

Here, $\hat{x}_i$ represents the predicted values, while $x_i$ represents the values obtained through the optimization procedure (COST, $CO_2$). Prediction models often face the challenge of overfitting. However, in this simple prediction model, the training and checking errors are comparable, indicating the absence of overfitting. It is important to note that the primary objective of this prediction model is to identify the inputs that exert the greatest influence on the output, rather than constructing a prediction model with minimal training error. To enhance the accuracy of the prediction model, it is advisable to incorporate more neurons in the neural networks. However, an increase in neurons may potentially lead to overfitting issues. The analysis also examines the combination of two inputs that hold the greatest influence over the output. The results of the parametric analysis unmistakably indicate that the total number of ESALs ($T_n$) is the most crucial parameter for achieving the

optimal cost of a pavement structure. Subsequently, the Marshall stability ($SM_{as} = SM_{ab}$), CBR of the subgrade ($CBR_{subgrade}$), and CBR of unbound layers ($CBR_{base} = CBR_{subbase}$) follow suit in terms of their significance.

Based on Figure 8, regarding the $CO_2$ emissions for the pavement structure, the bound layers (asphalt layers) are responsible for 96% of the $CO_2$ emissions, while the unbound layers account for the remaining 4%. This distribution of $CO_2$ emissions is valid for $T_n = 1 \times 10^8$, $CBR_{subgrade} = 7\%$, $CBR_{base} = 100\%$, and $SM_{as} = 2$ kN. The analysis shows that the fraction of $CO_2$ emissions caused by asphalt layers is much more sensitive to design parameters, while the fraction of pavement costs caused by asphalt layers is less sensitive to design parameters.

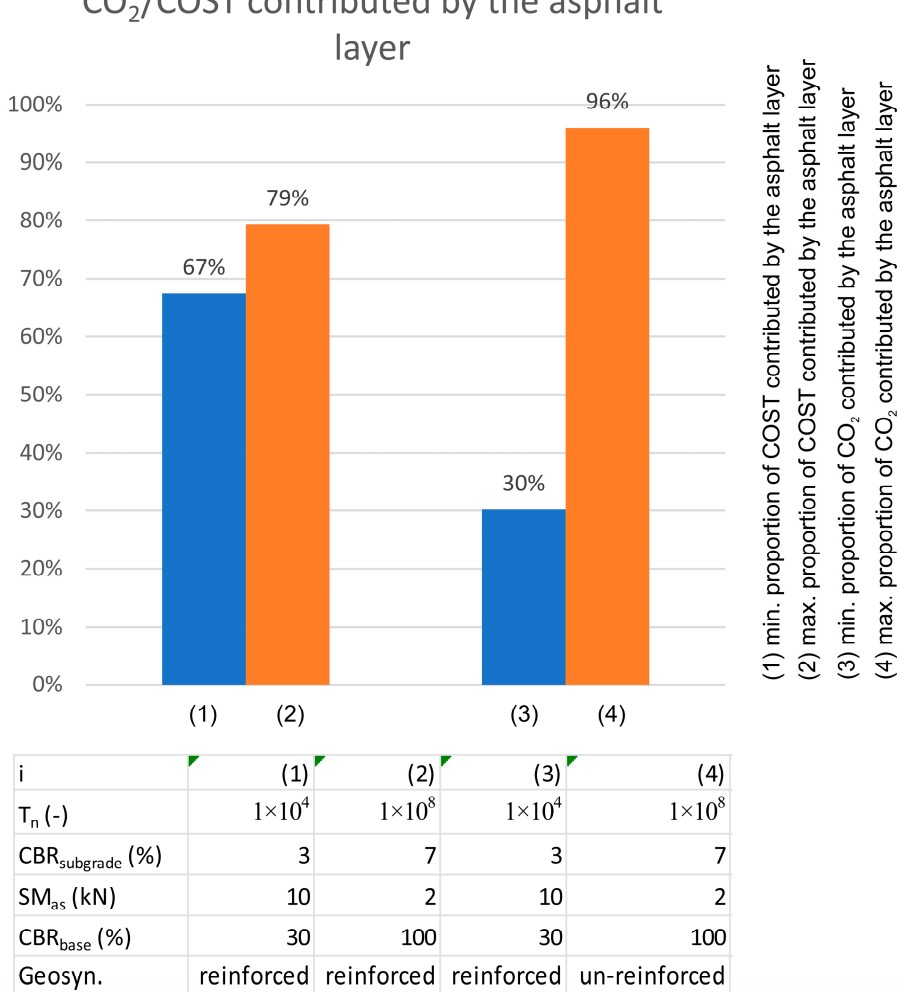

| i | (1) | (2) | (3) | (4) |
|---|---|---|---|---|
| $T_n$ (-) | $1 \times 10^4$ | $1 \times 10^8$ | $1 \times 10^4$ | $1 \times 10^8$ |
| $CBR_{subgrade}$ (%) | 3 | 7 | 3 | 7 |
| $SM_{as}$ (kN) | 10 | 2 | 10 | 2 |
| $CBR_{base}$ (%) | 30 | 100 | 30 | 100 |
| Geosyn. | reinforced | reinforced | reinforced | un-reinforced |

**Figure 8.** Proportion of COST and $CO_2$ emissions contributed by the asphalt layer.

## 5. Conclusions

The present work deals with the aspect of the material quality of the pavement structure, provided that other legal, organizational, and logistical conditions are also met. It examines how the incorporation of waste materials in bound and unbound pavement layers affects layer thicknesses and consequently on costs and $CO_2$ emissions. The inclusion of waste materials was accounted for via equivalence factors used in the empirical pavement design method. The geosynthetic reinforced and unreinforced pavement design was optimized for different traffic loads and material properties. The proportion of costs and $CO_2$ emissions of the asphalt layers were also calculated. The main conclusions are the following:

- For the most unfavorable design parameters examined in the parametric analysis, the thickness of the unbound layers increased from 91 cm to 120 cm (32% increase in thickness) when the CBR value of the base and sub-base layers decreased from 100% to 30%.
- For the most unfavorable design parameters examined in the parametric analysis, the thickness of the asphalt layer increased from 36 cm to 58 cm (61% increase in thickness) when the Marshall stability value of the asphalt layer decreased from 10 kN to 2 kN.
- The analysis shows that the proportion of $CO_2$ emissions caused by asphalt layers can vary from 30% to 96% depending on the design parameters, while the proportion of costs caused by asphalt layers only ranges from 67% to 79% for the same design parameters. This is due to the fact that the ratio of $CO_2$ emissions between the asphalt layer and the unbound layer is higher than the ratio of prices.
- The results of the parametric analysis show that the total number of ESALs ($T_n$) is the most important parameter for achieving the optimal cost of a pavement structure. This is followed by the Marshall stability (SM), the CBR value of the subgrade ($CBR_{subgrade}$), and the CBR value of the unbound layers ($CBR_{base} = CBR_{subbase}$) in terms of their importance.
- The use of geosynthetics could result in a 15% reduction in pavement structure cost and a 9% reduction in $CO_2$ emissions due to the reduced thickness of unbound layers. However, the use of geosynthetics could also result in an increase in road pavement structure cost and $CO_2$ emissions under favorable site conditions (e.g., with a CBR subgrade of 7%).
- The empirical design method for pavements limits the Marshall stability to approximately 10 kN, although the stability of asphalt concrete could be higher. Therefore, the mechanical-empirical design method could further improve the optimization model by considering even larger Marshall stability values.

The optimization model was developed in a general form that can provide an optimal solution for various design parameters including different traffic loads, site conditions, and material properties that depend on specific real project data. Further research is needed to evaluate how the properties of asphalt and unbound layers are altered by the addition of waste in various percentages and by the type of waste included. Once these relationships are known, waste reduction could also be determined in terms of cost and $CO_2$ emissions while achieving a reduction in waste deposition. Since this optimization model and, consequently, the results presented are based on an empirical pavement design method, further investigations could be investigated by semi-empirical pavement design methods or methods based on finite element modeling.

**Author Contributions:** Conceptualization, P.J., B.Ž., B.M., S.G. and C.G.; methodology, P.J., B.Ž., B.M., S.G. and B.M.; software, P.J. and R.V.; validation, P.J., B.Ž., B.M., S.G., C.G., R.V., T.B., Ş.Y., M.V.T. and B.E.K.; writing—original draft preparation, P.J., B.Ž., B.M., S.G. and C.G.; writing—review and editing, P.J., B.Ž., B.M., S.G., C.G., R.V., T.B., Ş.Y., M.V.T. and B.E.K. All authors have read and agreed to the published version of the manuscript.

**Funding:** This research was funded by the Slovenian Research Agency (ARIS) and Scientific and Technological research Council of Türkiye (TÜBİTAK) by supporting a bilateral project (grant numbers BI-TR/22-24-06 and 122N273) and by the EU project GEOLAB (grant number 101006512).

**Institutional Review Board Statement:** Not applicable.

**Informed Consent Statement:** Not applicable.

**Data Availability Statement:** The data presented in this study are available on request from the corresponding author.

**Conflicts of Interest:** The authors declare no conflict of interest.

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
