# Peer review of "Potential of Using Waste Materials in Flexible Pavement Structures Identified by Optimization Design Approach"

_sustainability, doi:10.3390/su151713141_

Round 1
Reviewer 1 Report
This paper explores the potential of incorporating waste materials into geosynthetic reinforced flexible pavement structures through an optimization design approach. The study analyzes the effects of waste materials on material properties, construction costs, and CO2 emissions. Geosynthetic reinforcement proves effective in pavements containing waste materials in unbound layers, reducing costs and CO2 emissions. The paper highlights the importance of waste reuse in road construction for sustainability and reducing environmental impact. The subject under instigation is of interest and is comprehensive, but lack of correct format and writing. More detailed problems are listed as follows:
1. Section “Abstract”, The abstract mentions that the optimal design of flexible pavements was achieved by minimizing construction costs, but it lacks specific details on the optimization process or any findings related to cost reduction. Adding a sentence or two on the cost optimization results would strengthen the abstract.
2. Section “Abstract”, While the abstract mentions the analysis of material properties and their effects on pavement structure costs and CO2 emissions, it would be helpful to include some quantitative data or conclusions to make the abstract more informative.
3. Section “1. Introduction”, While referencing existing studies on waste materials in asphalt pavements is commendable, providing more context on how this paper builds upon previous research and introduces novel aspects to the field would strengthen the introduction.
4. Section “1. Introduction”, The discussion on waste management barriers is informative, but it could be more focused and concise. Consider streamlining this section to highlight the key regulatory, organizational, logistical, and economic challenges related to waste reuse in pavement construction.
5. Section “1. Introduction”, While the use of a genetic algorithm for optimizing pavement structure cost and CO2 emissions is mentioned, the introduction does not elaborate on how this approach differs from previous optimization methods or why it was chosen for this study. A brief rationale for selecting the genetic algorithm should be included.
6. Section “4. Multi parametric optimization”, When discussing the results, consider providing more specific insights or practical implications based on the findings. For example, how can these optimal designs and parameter relationships be applied in real-world pavement projects to achieve cost and emissions savings?
7. Section “5. Conclusion”, It would be helpful to provide more context on the practical implications and applications of these conclusions in real-world pavement design and construction.
8. Section “5. Conclusion”, Overall, the conclusion effectively summarizes the study's main findings and implications. To enhance its impact, consider offering recommendations for future research directions to address the identified limitations.
9. Section “Reference”, Regarding the References section, there are some formatting issues that need to be addressed to ensure accuracy and consistency. Specifically, references 34, 35, and 36 are not formatted correctly. The citation style should be carefully checked and adjusted as necessary to adhere to the specified format guidelines.
Minor editing of English language required
Reviewer 2 Report
This study illustrates the design of a road paving with waste material.
The subject is particularly interesting and meets the needs of today's world to reduce environmental pollution. It is advisable to improve the introduction, especially to clarify the aim of the study, so that the reader is not confused or other expectations are raised. It is good to indicate from the beginning that a parametric analysis has been carried out.
It is desirable that the authors emphasise in the introduction and conclusions the importance of their work in relation to the current problems associated with climate change. Discussions need to be improved by emphasising the novelty of this study in comparison to other similar studies already found in the literature; a comparison is desirable.
Reviewer 3 Report
1. The problem significance needs to be enhanced in both Abstract and Introduction sections.
2. The intuition behind the proposed research needs to be clearly highlighted within the Introduction section.
3. Please ensure that all the keywords are mentioned within the Abstract text and are different from title to enhance the visibility.
4. The abstract section should include a bit of details about the quantitative results towards the end.
5. Add the limitations of the research.
Manuscript needs to be proofread by a native
Reviewer 4 Report
I found it interesting, and based on my opinion, it does not need revision.
Well written
Author Response
Please see the attachment. Thank you. The abstract and the introduction have been further improved.

Reviewer 5 Report
The research paper focuses on multi-parameter optimization model of flexible pavement structures with incorporated waste materials (plastic, fly ash, furnace dust etc.). The research is original and could be interesting for the readership of the Sustainability. However, the article is suffering from insufficient literature review, lack of comparison the obtained results with literature and validation of the modeling by the experimental results.
There are some concerns that need to be addressed:
1) In the introduction authors described lignin, car tires, glass and nylon bags as typical waste sources, however plastic remains the most abundant waste material. For this reason introducing plastic waste as binders in road asphalt production now is the most perspective way of reutilization plastic that could not be recycled. Moreover, in addition to scientific interests in the study of this problem, there is also a practical use of this approach – building “plastic roads” from plastic waste by well-known companies like MacRebur, VolkerWessels, TechniSoil Industrial etc. This fact should be highlighted.
2) The authors did not provide comprehensive literature review on the investigated topic, therefore there is lack of comparison the obtained results:
e.g.
10.1016/j.jmrt.2023.03.218
10.1007/s41939-023-00180-x
10.3390/polym15153293 and etc.
Also, preferably authors should try to find the experimental results that corelates or could validate their modelling.
3) On the Figure 3 it is shown that for bitumen mixture contained waste material from PE carry bags Marshall stability could be in the range 12-16 kN for 2.5-10 % of waste content. However, during modeling only mixtures with Marshall stability in the range 2-10 kN were considered (Figure 5) – could you comment this.
4) The paragraph lines 362-396 is hard to read could you restructure and divide it on subparagraphs.
5) Quality of all graphics should be improved (not less than 300 dpi).
Round 2
Reviewer 1 Report
My comments have been addressed. It is now acceptable.
Fine.
Reviewer 5 Report
I satisfied with the authors responses. The manuscript was improved during the revision.
I have several comments:
1) For Table 2 please provide the necessary number of significant digits for each value. Could you comment why the carbon index and thickness of different layers have different number of significant digits?
2) In the Tables 4 and 5 I suggest use m for layer thickness instead of cm.
3) Please correct the subscript in the formula on the lines 133 and 417.
